# The Role of Passion and Achievement Goals in Optimal Functioning in Sports

**DOI:** 10.3390/ijerph18179023

**Published:** 2021-08-28

**Authors:** Jany St-Cyr, Robert J. Vallerand, Léandre Alexis Chénard-Poirier

**Affiliations:** Laboratoire de Recherche sur le Comportement Social, Département de Psychologie, Université du Québec à Montréal, 100 rue Sherbrooke Ouest, P.O. Box 8888 Centre-Ville, Montréal, QC H3C 3P8, Canada; vallerand.bob@gmail.com (R.J.V.); leandre.chenard.poirier@gmail.com (L.A.C.-P.)

**Keywords:** passion, harmonious passion, obsessive passion, sport, optimal functioning in society, well-being, achievement goals

## Abstract

This study aimed to test the role of passion in the cognitive goals pursued in sport and the level of Optimal Functioning in Society (OFIS) derived from such sport engagement. A total of 184 competitive water polo and synchronized swimming athletes completed a questionnaire assessing their passion for their sport, achievement goals, and various scales assessing their level of OFIS (e.g., subjective well-being, relationship with their coach, sport performance, and intentions to continue in sport). It was hypothesized that harmonious passion (HP) would be positively associated with mastery goals while obsessive passion (OP) would be positively associated with mastery, performance-approach, and performance-avoidance goals. In turn, mastery goals were expected to positively lead to the four components of OFIS, whereas performance-approach and performance-avoidance goals should display less adaptive relationships with OFIS. The results of a path analysis generally supported the proposed model. As hypothesized, these findings suggest that HP leads to a more adaptive cognitive engagement in sport (than OP) that, in turn, fosters higher levels of optimal functioning.

## 1. Introduction

Much research reveals that physical activity and sport can contribute to one’s well-being [1,2,3]. However, such a contribution may not be automatic. Indeed, research also reveals that how one engages in sport can affect the cognitive goals that are pursued and these, in turn, can determine the benefits that one derives from such engagement. One factor that has been found to contribute to cognitions and well-being is the concept of passion [4]. Passion for sport and physical activity can enhance the quality of one’s cognitive engagement and adaptive cognitions, contributing to optimal well-being. However, some forms of passion can also lead to maladaptive cognitions and less-than-optimal well-being. The present research empirically tests these issues.

The Dualistic Model of Passion (DMP) [4,5] defines passion as a strong inclination toward an activity that people like (or even love), find important, in which they invest time and energy, and which is part of their identity. The DMP further posits the existence of two types of passion, each associated with different processes and outcomes. Harmonious passion (HP) is the result of an autonomous internalization of the activity in identity. Specifically, for athletes with an HP, their sport is part of who they are, and they practice it only out of love for this activity. HP mainly leads to adaptive processes and outcomes because people fully engage in the activity they enjoy while maintaining a balance with other aspects of their life. For HP, examples of adaptive processes and outcomes are, respectively, deliberate practice, mindfulness, and task coping, as well as performance, concentration, flow, positive emotions, and well-being [4,6,7]. Obsessive passion (OP), on the other hand, originates from a controlled internalization of the beloved activity in identity. Specifically, when athletes have an OP, the sport that they love becomes part of who they are along with external contingencies such as social recognition. This leads to the adoption of a compulsive engagement in their sports, which in turn, leads to less adaptive processes and outcomes such as avoidance coping, rumination, negative emotions, and the adoption of rigid and risky behaviors. Over the past 20 years or so, hundreds of studies have provided support for the DMP in a number of areas [4,6,8,9] including sport and exercise [7,10,11].

Research has shown that HP and OP relate differently to the cognitive goals pursued during one’s sport engagement [12,13]. According to Elliot’s trichotomous framework [14,15], when one engages in a meaningful activity, three fundamental types of cognitive goals can be pursued: mastery (pursuing the goal of personal improvement), performance approach (trying to be better than others), and performance avoidance (avoiding being worse than others). Research reveals that HP facilitates the adoption of mastery goals whereas OP leads to some mastery goals, but mainly to performance-approach and performance-avoidance goals. These findings have been obtained in a number of sport and exercise studies [12,13,16]. In addition, mastery goals have been found to lead to adaptive outcomes such as performance, well-being, and high-quality relationships, whereas performance-avoidance goals typically have negative relationships with performance and well-being [15]. Avoidance goals have been typically shown to be associated with worries or intrusive thoughts related to the possibility of failure, which have the consequence of decreasing one’s mental focus and, undermining one’s performance [17]. Finally, performance-approach goals are generally positively associated with performance [18] but results are inconsistent with regard to their relationship with well-being and positive outcomes due to feelings of obligation to outperform others which can trigger negative experiences [19].

A person’s investment in pursuing their goals can influence the outcomes and, therefore, the level of functioning achieved. Vallerand [20,21] has proposed the concept of Optimal Functioning in Society (OFIS). The concept of OFIS refers to high levels of psychological, physical, and relational well-being, high performance in one’s main field of endeavor, and contribution to society. Such high level of functioning can be reached when fully engaging in an activity out of HP, which enables positive experiences and promotes adaptive results and self-growth [22]. Such results should not be observed with OP since they lead to fewer positive, and even to some negative, experiences. Recent research has supported the adaptive role of HP in OFIS and the less adaptive and, at times, the maladaptive role of OP in OFIS [23]. However, the role of OFIS has yet to be studied in sport and physical activity.

In line with the above, the purpose of the present study was to test the role of HP and OP in the adoption of the three cognitive goals and in the promotion of OFIS; no study so far has done so in the realm of sport and physical activity. More specifically, the purpose of the present research was to test an integrated model involving passion, cognitive goals, and OFIS with high-level sport participants who were members of their respective junior national team of water polo and synchronized swimming. Because participants were relatively young (teenagers), only the first four elements of OFIS (psychological, physical, and relational well-being, and sport performance) and not contribution to society, were assessed. Furthermore, since the participants were already all very physically active, physical health as such was not assessed. Rather, intentions to continue engaging in sport were used as an indirect indicator (proxy) of maintaining one’s physical health. Indeed, research has shown that maintaining physical activity practices can have many benefits throughout life [24]. Overall, it was hypothesized that HP should foster mastery goals, whereas OP should be positively associated with all three goals. In turn, mastery goals should be positively associated with all four OFIS elements, whereas performance-approach and performance-avoidance goals should display less adaptive relationships with the OFIS elements.

## 2. Materials and Methods

### 2.1. Participants and Procedure

A total of 192 respondents completed the survey. There were 79 male and 113 female participants, all competitive water polo and synchronized swimming athletes. These athletes were, on average, 15.80 years old (S.D. = 4.15 years). They completed a survey on a voluntary basis. The questionnaire was completed early in their competitive season. They were high-level athletes on the junior national team of their respective sport. Athletes had been practicing their sport for an average of 4.55 years (S.D. = 4.15 years). They reported playing their sport about 4.68 times per week (S.D. = 1.62) for an average duration of 131.83 min each time (S.D. = 105.93 min).

### 2.2. Measures

For the entire self-report questionnaire, the scales referred to the sport that participants engaged in (i.e., synchronized swimming or water polo). Additionally, for all measures, a 7-point Likert scale was used (1 = *not agree at all* to 7 = *very strongly agree*). All measured items are available in the Appendix A.

#### 2.2.1. Passion

The Passion Scale [5,25] was used to assess HP (e.g., “Synchronized swimming/water polo is in harmony with other activities in my life”) and OP (e.g., “I cannot imagine my life without synchronized swimming/water polo”) for the sporting activity. The HP and OP subscales were measured seven items each [5]. The Passion Scale showed high levels of validity and reliability. More than 20 studies supported the Passion Scale and its psychometric properties, factorial validity and its invariance for gender, age, and the activity performed, as well as its reliability [25,26]. In the present study, Cronbach alpha coefficients of α = 0.75 and α = 0.86 were obtained, respectively, for the HP and OP subscales.

#### 2.2.2. Achievement Goals

To assess achievement goals, the 18-item scale from Elliot and Church [14] was used and adapted to the sport’s domain. The scale contains three subscales of six items: mastery (α = 0.81; e.g., “*I want to develop my skills as much as possible*”), performance-approach (α = 0.83; e.g., “*I am motivated by the thought of outperforming my teammates*”), and performance-avoidance goals (α = 0.78; e.g., “*I just want to avoid playing poorly*”). This scale consistently displayed high levels of reliability and validity [14].

#### 2.2.3. Coach-Athlete Relationship

The relational well-being component of OFIS was measured with five items assessing the quality of the coach-athlete relationship. These items asked athletes to identify at what level they felt *supported*, *confident*, *understood*, *listened to* and *valued* by their coach (α = 0.84). This scale was adapted from the Quality of Relationship Scale which consistently displayed high levels of reliability and validity [27].

#### 2.2.4. Performance

Considering that the survey was completed at the start of the season, a combination of different scales was used to create a self-perceived measure of performance (an element of OFIS). These scales were all reliable and valid, and included two items assessing expected performance (e.g., “*I expect to do well this season*”); two items from the confidence and achievement motivation subscale (e.g., “*I get the most out of my talent and my skills*”); four items from the peaking under pressure subscale (e.g., “*I make fewer mistakes when the pressure is on because I am concentrating better*”), both from the Athletic Skills Coping Inventory [28]; and four items inspired by the General Self-Efficacy Scale [29] (e.g., “*I feel efficient in the different aspects of synchronized swimming/water polo*”). Overall, the Cronbach alpha coefficient was 0.84.

#### 2.2.5. Subjective Well-Being

The Satisfaction with Life Scale [30] was used to assess the psychological well-being element of OFIS. This scale was made up of five items measuring the level of satisfaction with one’s life in general (α = 0.77; e.g., “*I am satisfied with my life*”). This scale was used in various cultural contexts and with various populations and showed high levels of reliability and validity [30].

#### 2.2.6. Intentions

The intentions to continue sport engagement were an indirect indicator of the OFIS physical well-being variable and were measured with four items (α = 0.82; e.g., “*I will continue to do synchronized swimming/water polo*”). These items were inspired by previous research on intentions to continue physical exercise [31].

### 2.3. Statistical Analysis

#### 2.3.1. Preliminary Analyses

Preliminary analyses were conducted using IBM SPSS Statistic (Version 25) software. Data distribution and outliers were examined. All skewness indices were between −1 and 1, which assumes a normal distribution of the data, except for the mastery goals. Thus, this variable was squared to distribute the data normally. Then, the examination of box plots revealed no univariate outlier. No multicollinearity (VIF < 5) between variables was revealed. However, Mahalanobis distances with a critical chi-square value at *p* = 0.05 revealed the presence of 16 multivariate outliers. Further analyses revealed that only the first eight outliers negatively affected the data. In order to retain as many participants as possible for statistical purposes, only these eight outliers were eliminated. The Z scores of each variable were calculated.

#### 2.3.2. Main Analyses

Path analyses were conducted on Mplus 8.4 [32] using maximum likelihood estimation. The Root Mean Square Error of Approximation (RMSEA) index, the Comparative Fit Index (CFI), and the Tucker–Lewis Index (TLI) were used as fit indices to assess the statistical adequacy of the models that were tested. According to Kline [33], RMSEA must be less than 0.08, and CFI and TLI must be at least greater than 0.90 in order support the statistical adequacy of a model. The low level of missing responses (0% to 0.54%) was handled using *full information maximum likelihood* [34].

## 3. Results

Descriptive statistics (the means and standard deviations of each scale), and bivariate correlations between variables were conducted. The results are shown in Table 1.

The main analyses focused on testing the hypothesized model. This model posited that HP should be positively related to mastery goals, whereas OP should be positively related to mastery, performance-approach, and performance-avoidance goals. In turn, mastery goals were expected to be positively related to the coach-athlete relationship, performance, subjective well-being, and intentions to continue sport engagement. On the other hand, performance-approach goals were expected to be positively related to all dependent variables, but less so than mastery goals. Finally, performance-avoidance goals were hypothesized to be negatively related to all outcomes.

Results indicated that the hypothesized model did not yield an adequate fit to the data. Therefore, in line with Kline [33], non-significant paths were removed. This later model yielded an adequate fit to the data: χ^2^ (15) = 27.701, *p* = 0.024; RMSEA = 0.068 (0.025; 0.107); CFI = 0.958; TLI = 0.901. Results appear in Figure 1 and showed that HP was positively associated with mastery goals (β = 0.343, *p* < 0.001), whereas OP was positively associated with mastery (β = 0.230, *p* = 0.006), performance-approach (β = 0.259, *p* = 0.001), and performance-avoidance goals (β = 0.261, *p* < 0.001). In turn, mastery had positive and significant links with each of the four outcomes, namely: coach-athlete relationship (β = 0.236, *p* = 0.001), performance (β = 0.499, *p* < 0.001), subjective well-being (β = 0.376, *p* < 0.001), and intentions to continue sport involvement (β = 0.415, *p* < 0.001). Performance-approach goals were only significantly and positively associated with performance (β = 0.199, *p* = 0.009). Finally, performance-avoidance goals were negatively related to performance (β = −0.219, *p* = 0.005) but only tangentially and negatively associated with subjective well-being (β = −0.138, *p* = 0.055).

Further, the statistical significance of indirect effects present in this mediation model was tested using 95% bias-corrected bootstrap confidence intervals based on 10,000 resamples. Mastery goals significantly mediated the relationships between HP and all four outcome variables: coach-athlete relationship (B = 0.080; *SE* = 0.033; 95% CI = 0.028, 0.160), performance (B = 0.178; *SE* = 0.048; 95% CI = 0.092, 0.278), subjective well-being (B = 0.134; *SE* = 0.042; 95% CI = 0.063, 0.226), and intentions to continue sport participation (B = 0.153; *SE* = 0.045; 95% CI = 0.076, 0.252). Mastery goals also significantly mediated the relationships between OP and coach-athlete relationship (B = 0.050; *SE* = 0.025; 95% CI = 0.013, 0.114), performance (B = 0.112; *SE* = 0.046; 95% CI = 0.030, 0.214), subjective well-being (B = 0.084; *SE* = 0.035; 95% CI = 0.025, 0.166), and intentions to continue sport (B = 0.096; *SE* = 0.039; 95% CI = 0.027, 0.182). In addition, performance-approach goals significantly mediated the relationship between OP and performance (B = 0.050; *SE* = 0.026; 95% CI = 0.012, 0.117). Finally, the performance-avoidance goals significantly mediated the relationship between OP and performance (β = −0.056; *SE* = 0.025 95% CI = −0.119, −0.017).

## 4. Discussion

The aim of the present study was to test the role of HP and OP in cognitive goals when engaged in sport and OFIS. It was hypothesized that HP would foster mastery goals and that OP should be positively associated with mastery, performance-approach, and performance-avoidance goals. In turn, mastery goals were expected to be positively associated with all four OFIS elements: the coach-athlete relationship, performance, subjective well-being, and intentions to continue sport engagement. In addition, it was hypothesized that performance-approach and performance-avoidance goals should display less adaptive relationships with the OFIS elements. Overall, the results of the path analysis supported the hypothesized model. These findings had a number of implications for research as well as for a better understanding of athletes’ adaptive functioning in sport.

The first implication was that the present findings replicated the links between passion and achievement goals. In line with past research conducted in sports and other fields [12,13,35], the relationships between HP and mastery goals and those between OP and mastery, performance-approach, and performance-avoidance goals were reproduced. These results were consistent with previous research which showed that HP fosters an adaptive achievement process focused on self-growth while OP was less optimal and triggers a more adversarial process where one tries to achieve goals in multiple ways, including trying to “beat” one’s teammates [12,13,35]. Thus, not all types of engagement in sports led to the adoption of an adaptive cognitive set. Rather, HP facilitated a harmonious road to sport engagement that fostered self-growth and mastery goals that should be pursued. Conversely, OP led to a more convoluted road to sport engagement where some self-growth is sought but mostly a mix of approach and avoidance performance-oriented goals were pursued with less than optimal benefits. Thus, one’s passion (HP or OP) determined the cognitive goals that were pursued when engaging in sports.

A second implication is that goals mediate the effect of passion on OFIS. Thus, passion helps to understand the quality of one’s cognitive involvement in sport (the goals) and how this cognitive involvement leads to health and well-being outcomes. Mastery goals facilitate coach-athlete relationship, performance, subjective well-being, and intentions to continue sport engagement and thus maintain one’s health. A recent study among dancers has also demonstrated the positive role of mastery goals in the quality of relationships with dance partners [16]. This particular study showed that HP positively predicted mastery goals which, in turn, predicted better relationships with one’s dance partners and the dance community at large. These findings are very similar to those obtained in the present study where HP positively predicted mastery goals which, in turn, predicted a better coach-athlete relationship. This positive relationship between coaches and athletes can be explained by the fact that athletes with mastery goals are more inclined to focus on *how* they can improve. As the coach is an expert who can contribute to the athletes’ development and improvement, maintaining a good relationship with the coach can facilitate such self-growth. In addition, mastery goals do not limit growth to performance but also apply to all dimensions of the self. Thus, with HP, one pursues mastery goals that also foster positive psychological well-being, health, and relationships without neglecting performance [36,37,38]. Within this context, positive experiences in sport are known to promote enjoyment which is a determining factor in the sustained participation in sport [39,40]. Moreover, the recommendations of an expert panel on why children and young people drop out of sports affirm that in order to encourage young people to continue their sport involvement, success must be defined not in terms of victory, but rather in terms of positive participation which allows the development and mastery of skills [41]. On the other hand, the two other types of goals are more limited in their effect with performance-avoidance goals being maladaptive pertaining to performance, and performance-approach goals only leading to performance. These mediating roles of performance-approach and performance-avoidance goals between OP and performance replicate the results from Vallerand et al. [13]. Other research [42] has shown a negative association between performance-avoidance goals and well-being, but this relationship was not significant in the present study. Future research should try to replicate this last result.

A third implication is that, overall, HP leads to higher levels of optimal functioning than OP. Indeed, results from the indirect effects revealed that HP was positively associated with all four OFIS elements through its relationship with mastery goals. Therefore, having an HP for one’s sport seems to pave the way for a healthier engagement where one experiences a more complete form of adaptive well-being comprised of satisfaction with life, a positive relationship with one’s coach, intentions to pursue healthy sport engagement, while achieving high performance in the process. Conversely, OP leads to lower levels of functioning. Indeed, although OP leads to some positive outcomes through mastery and performance-approach goals, its association with performance-avoidance goals also leads to lower overall benefits than HP. Once more, these findings reinforce the fact that two ways for sport involvement emerge. Firstly, HP leads one to focus on self-improvement and self-growth [22]. This focus causes athletes to have a healthier form of involvement in the context of their sport and to derive higher levels of adaptive functioning. Secondly, triggered by an OP, athletes are motivated to focus on beating others and leads them to have more mixed, or even negative, experiences in the context of their sport. These findings are in line with past research on passion and OFIS [23] and are the first to focus specifically on OFIS in sport.

Some limitations of the present study should be mentioned. Firstly, all variables were self-reported, thus objective measures should be used in future research. Secondly, a cross-sectional design was used. It is recommended that an experimental design is used in future research so that the causal role of passion in cognitive goals and OFIS is replicated. Thirdly, participants’ contribution to society, the last element of OFIS, was not assessed as it was felt that participants were somewhat too young to assess this dimension. Future research in sport should do so when using older sport participants.

## 5. Conclusions

In summary, the present study is the first to assess the role of passion in OFIS in the sport context. The results shed light on the two types of passion in sport engagement and their effects on cognitive goals and OFIS. The first and more harmonious road has more adaptive effects through its relationship with mastery goals. Such goals lead athletes to experience OFIS comprised of high levels of psychological and relational well-being, high performance, and intentions to continue sport involvement, and thus maintain health. On the other hand, the second and more obsessive road to sport engagement leads to adopting less adaptive cognitive goals that include performance-approach and performance-avoidance goals. These goals, in turn, foster more limited functioning. Overall, these findings suggest that simply “doing” sport is not enough. To fully derive well-being benefits from sport engagement, one should have an HP for sport as it fosters a more adaptive cognitive engagement (mastery goals) in sport than an OP that, in turn, fosters higher levels of well-being and optimal functioning.

## Figures and Tables

**Figure 1 ijerph-18-09023-f001:**
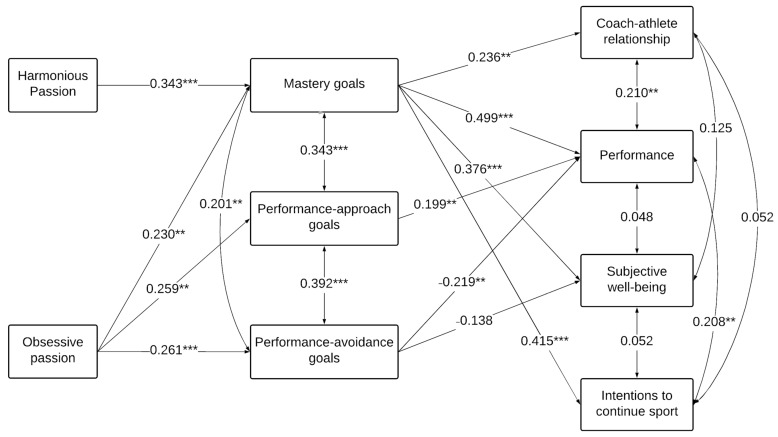
Results of path analyses showing the relations between passion, achievement goals and OFIS components. *N* = 184; ** *p* < 0.01, *** *p* < 0.001. Standardized betas are presented.

**Table 1 ijerph-18-09023-t001:** Means, Standard Deviation, and Correlations.

	M	S.D.	1	2	3	4	5	6	7	8	9
1. Harmonious passion	5.546	0.911	-	0.726 **	0.517 **	0.232 **	0.148 *	0.304 **	0.382 **	0.295 **	0.325 **
2. Obsessive passion	4.219	1.356		-	0.477 **	0.259 **	0.261 **	0.190 **	0.347 **	0.185 *	0.329 **
3. Mastery goals	6.111	0.826			-	0.418 **	0.274 **	0.237 **	0.522 **	0.338 **	0.416 **
4. Performance-approach goals	4.595	1.310				-	0.433 **	0.097	0.303 **	0.183 *	0.118
5. Performance-avoidance goals	4.349	1.259					-	0.004	−0.006	−0.042	0.114
6. Coach-athlete relationship	5.368	0.977						-	0.304 **	0.202 **	0.144
7. Performance	5.191	0.845							-	0.253 **	0.362 **
8. Subjective well-being	5.234	0.993								-	0.185 *
9. Intentions to continue sport	4.969	1.399									-

Note. *N* = 184; M = mean; S.D. = standard deviation; * *p* < 0.05, ** *p* < 0.01.

## Data Availability

The raw data supporting the conclusion of this article is available upon request from the corresponding author.

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
