# Peer review of "The Role of Passion and Achievement Goals in Optimal Functioning in Sports"

_ijerph, 2021, doi:10.3390/ijerph18179023_

Round 1

Reviewer 1 Report

This is an excellent article.  It is written well, and the science seems very sound.  I have the following minor suggestions:

  1. Line 188, please change don't to do not (I prefer formal English)
  2. Perhaps the word effect is a better choice than impact. Again, I prefer formal English in published articles. This occurs just a few times in the article and is corrected easily with the search-and-replace function.
  3. In Table 1, perhaps aligning the numbered items (e.g., 1. Harmonious passion) with an indented left margin would look better than having the items centered.
  4. Lines 250-251, perhaps the authors could add another sentence or two to expand on the mastery goals with respect to dancers.  That is an interesting part of the discussion, and a little more information may be helpful to the reader. Dancers and athletes can be very similar. 

I would recommend publishing the article with the minor changes mentioned above. 

Author Response

Response to Reviewer 1 Comments

Thank you for the positive feedback. It is highly appreciated.

Point 1: Line 188, please change don't to do not (I prefer formal English).

Response 1: Changes were made throughout the manuscript accordingly.

Point 2: Perhaps the word effect is a better choice than impact. Again, I prefer formal English in published articles. This occurs just a few times in the article and is corrected easily with the search-and-replace function.

Response 2: We agree with the Reviewer. Thus, changes were made throughout the manuscript accordingly.

Point 3: In Table 1, perhaps aligning the numbered items (e.g., 1. Harmonious passion) with an indented left margin would look better than having the items centered.

Response 3: We agree that the items should be aligned. Thus, items were properly aligned in Table 1.

Point 4: Lines 250-251, perhaps the authors could add another sentence or two to expand on the mastery goals with respect to dancers. That is an interesting part of the discussion, and a little more information may be helpful to the reader. Dancers and athletes can be very similar.

Response 4: We agree with the reviewer that more information on the role of mastery goals with respect to the relationships of dancers (Guilbault et al., 2020) may be helpful and may strengthen the discussion. We have therefore added more information on this issue (see p.7, lines 262-268).

Reviewer 2 Report

  1. Page 1, line 29: Why consider paper’s title as the title of part one?

  1. All variables in this survey were measured by self-report questionnaire. The data would be reliable and valid when measured items/scale have good reliability and validity. In the section of Measures, I only find Cronbach alpha coefficients and source of scale. What about test-retest reliability and validity?

  1. The description of the measured item was relatively short and unclear. Considering that there are so many variables involved in this study, it is necessary to show the measured items in Supplementary Materials. And I suggest the author add relevant description of measured items in Measures section.

  1. Page 3, line 109: I did not find the description of statistical method in Materials and Methods. This section in this type of the paper is important. I suggest moving the the description of the statistical methods in the Results section to the Methods section.

  1. Page 5, line 204: what is the title of Figure 1.? And the note of the figure should be under the title.

  1. References must be numbered in order of appearance in the text, and reference numbers should be placed in square brackets. For example, line of 31 and 40, page 1; line of 50 and 58, page 2. Please standardize your reference list according to Instructions for authors (Page 8, line 322).

Author Response

Response to Reviewer 2 Comments

Point 1: Page 1, line 29: Why consider paper’s title as the title of part one?

Response 1: We have changed the title of this section and named it “Introduction” as well as numbered each section as asked in the Instructions for Authors.

Point 2: All variables in this survey were measured by self-report questionnaire. The data would be reliable and valid when measured items/scale have good reliability and validity. In the section of Measures, I only find Cronbach alpha coefficients and source of scale. What about test-retest reliability and validity?

Response 2: All of the scales used in this study have shown high levels of validity and reliability in past research. In line with the Reviewer’s comment, we have provided additional information regarding the reliability and validity for each scale in the Measures section.

Point 3: The description of the measured item was relatively short and unclear. Considering that there are so many variables involved in this study, it is necessary to show the measured items in Supplementary Materials. And I suggest the author add relevant description of measured items in Measures section.

Response 3: With respect to point # 2, we have provided additional information to better describe the items measured by each scale. In addition, we’ve included the measured items in the Supplementary Materials as suggested.

Point 4: Page 3, line 109: I did not find the description of statistical method in Materials and Methods. This section in this type of the paper is important. I suggest moving the description of the statistical methods in the Results section to the Methods section.

Response 4: As suggested by the Reviewer, we have moved the description of the statistical methods from the Results section to the Methods section (see changes on p. 4).

Point 5: Page 5, line 204: what is the title of Figure 1.? And the note of the figure should be under the title.

Response 5: We thank Reviewer 2 for this astute observation. We have added the title of Figure 1 and made the appropriate changes.

Point 6: References must be numbered in order of appearance in the text, and reference numbers should be placed in square brackets. For example, line of 31 and 40, page 1; line of 50 and 58, page 2. Please standardize your reference list according to Instructions for authors (Page 8, line 322).

Response 6: We thank Reviewer 2 for this comment. We have made changes accordingly throughout the manuscript.

Reviewer 3 Report

The paper address an important issue and the construction of the argument is solid

The methodology is fine and the analysis is well conducted.

The conclusions could be expanded on the basis of the existing literature

Author Response

Response to Reviewer 3 Comments

Point 1: The paper addresses an important issue, and the construction of the argument is solid.

Response 1: We wish to thank Reviewer 3 for his/her positive comment regarding the paper.

Point 2: The methodology is fine, and the analysis is well conducted.

Response 2: We thank Reviewer 3 for this kind comment.

Point 3: The conclusions could be expanded on the basis of the existing literature.

Response 3: We agree that the conclusions could be expanded based on the existing literature. We now provide more information in the Conclusions section (see changes on p.8).

Round 2

Reviewer 2 Report

Thank you for addressing my concerns. Best wishes!